# Hidden Truth in Cerebral Concussion—Traumatic Axonal Injury: A Narrative Mini-Review

**DOI:** 10.3390/healthcare10050931

**Published:** 2022-05-18

**Authors:** Sung Ho Jang, Dong Hyun Byun

**Affiliations:** Department of Physical Medicine and Rehabilitation, College of Medicine, Yeungnam University, NamKu, Taegu 42415, Korea; strokerehab@hanmail.net

**Keywords:** concussion, mild traumatic brain injury, axonal injury, diffusion tensor imaging, diffusion tensor tractography

## Abstract

This study reviewed traumatic axonal injury (TAI) in patients with concussion. Concussion refers to transient changes in the neurological function of the brain resulting from head trauma that should not involve any organic brain injury. On the other hand, TAI has been reported in autopsy studies of the human brain and histopathological studies of animal brains following concussion before the development of diffusion tensor imaging (DTI). The diagnosis of TAI in live patients with concussion is limited because of the low resolution of conventional brain magnetic resonance imaging. Since the first study by Arfanakis et al. in 2002, several hundred studies have reported TAI in patients with concussion using DTI. Furthermore, dozens of studies have demonstrated TAI using diffusion tensor tractography for various neural tracts in individual patients with concussion. Hence, DTI provides valuable data for the diagnosis of TAI in patients with concussion. Nevertheless, the confirmation of TAI in live patients with concussion can be limited because a histopathological study via a brain biopsy is required to confirm TAI. Accordingly, further studies for a diagnostic approach to TAI using DTI without a histopathological test in individual patients with concussion will be necessary in the clinical field.

## 1. Introduction

Cerebral concussion, a type of traumatic brain injury (TBI), is a transient reversible neurologic dysfunction resulting from an external mechanical force transmitted to the brain through a physical impact on the head [1,2,3]. Concussion generally lasts for less than 24 h, and recovery usually occurs within two–three weeks after head trauma [1,2]. In other words, concussion refers to transient changes in the neurological function of the brain resulting from an acute head trauma that does not involve an organic brain injury [1,2]. The term mild TBI (mTBI) refers to a traumatic brain injury with a loss of consciousness (LOC) for less than 30 min and has been used interchangeably with the term concussion [4]. Concussion and mTBI are generally not accompanied by brain lesions detectable with conventional brain magnetic resonance imaging (MRI) [1,2]. Concussion has been used since the 10th century, whereas mTBI has been used since 1993, when it was defined by the American Congress of Rehabilitation Medicine [4,5]. mTBI is used mainly as the academic terminology, whereas concussion has been used more commonly in the clinical field and community (Table 1).

Traumatic axonal injury (TAI) indicates the tearing of axons by indirect shearing forces during acceleration, deceleration, and rotation of the brain or direct head trauma [8,9,10,11,12,13]. Since the 1960s, histopathological studies after autopsy have revealed TAI in patients with concussion who died from other causes [14,15,16]. However, the diagnosis of TAI in live patients with concussion is limited because conventional brain MRI is not sensitive to detecting TAI in concussion. Since the introduction of diffusion tensor imaging (DTI), many researchers have demonstrated TAI in live patients with concussion or mild TBI since the 2000s [8,17,18,19,20,21,22,23,24,25,26,27,28,29,30,31]. Recent DTI-based studies have reported that TAI was more severe in concussion than in mTBI [32,33]. Hence, this study reviewed TAI focusing on concussion, which included the concept of mTBI in terms of the severities of LOC and TAI.

This study reviewed TAI in concussion in terms of ambiguous diagnostic criteria for concussion, the limitations of conventional brain MRI in the detection of TAI, the history of TAI, and the demonstration of TAI using DTI.

## 2. Ambiguity of Diagnostic Criteria for Concussion and Diffuse Axonal Injury

Head trauma is defined as concussion when LOC following a head trauma lasts less than six hours but it is defined as a diffuse axonal injury when LOC lasts six hours or more [6,34]. On the other hand, the six-hour LOC criterion to distinguish between concussion and diffuse axonal injury is deficient of convincing evidence because TAI lesions have been detected in concussion [35,36,37]. Gennarelli’s study, which has been frequently cited as the reference of this criterion, did not include a reference for the origin of the criterion [34], but another study by Gennarelli et al. reported that approximately half (46.7%) of monkeys studied that exhibited LOC for less than six hours after whiplash had axonal injury lesions on a histopathological test [35]. Other studies detected axonal injury lesions through autopsy in patients with concussion or mild TBI who died of other diseases [14,15,16]. Furthermore, axonal injury lesions were detected in conventional brain MRI in 12.5–30% of patients with mTBI; Mittl et al. (1994) and Topal et al. (2008) detected diffuse axonal injury lesions in conventional brain MRI in 30% and 12.5%, respectively [36,37].

## 3. Limitation of Conventional Brain MRI in the Detection of TAI in Concussion

In individual patients with concussion, the absence of neural injury has been determined mainly based on normal findings in conventional brain MRI. However, the low resolution of conventional brain MRI compared to the neuron number of the human brain can be problematic. In detail, there are at least ten billion neurons in the human brain, and a brain MRI scan consists of approximately 300,000 voxels (a voxel refers to a point on a three-dimensional plane) [6,8]. Based on these values, each voxel within a brain MRI scan would indicate the status of at least approximately 30,000 neurons. This low resolution of brain MRI makes it difficult for a brain MRI to accurately reflect the state of neural injury. Thus, normal findings in a conventional brain MRI may not indicate whether the brain is in a normal state [6,8]. Because of this limitation of conventional brain MRI, only approximately 50% of axonal injury lesions can be detected, even in patients with diffuse axonal injury [38,39,40,41,42].

## 4. Evidence of Organic Brain Injury in Concussion

Recovery from concussion should be complete without sequelae because it is a transient dysfunction of the brain [1,2]. However, a significant proportion of patients with concussion show sequelae three months after onset. These patients were diagnosed with post-concussion syndrome, which has been recognized as a psychological problem [1,2,43,44,45]. Rutherford et al. reported that approximately 15% of patients with concussion showed sequelae one year after concussion [45]. In contrast, McMahon et al. reported that one year after concussion, 82% of patients had at least one symptom of post-concussion syndrome [46]. After introducing DTI, many studies showed that TAI is the underlying pathophysiology of post-concussion syndrome [19,47]. On the other hand, opinions critical of the terms concussion and mTBI, which indicate a benign condition, have been suggested. Some researchers suggested that the terms concussion and mTBI should be avoided because many of these patients show severe sequelae and exhibit heterogeneous clinical features, ranging from mild to severe [46,48,49,50].

Many studies have provided evidence of organic brain injuries using various evaluation methods, such as single-photon emission computed tomography, positron emission tomography, evoked potential testing, electroencephalography, functional MRI, magnetoencephalography, and blood biomarkers [51,52,53,54]. However, these evaluation methods have limitations because none can show the definite pathology or localization of a lesion at the subcortical white matter level.

## 5. History of Traumatic Axonal Injury in Concussion

TAI is a pathological diagnostic terminology. Thus, a histopathological study via a brain biopsy is required to confirm the TAI of a neural structure in patients with concussion [8]. Conversely, brain biopsy in patients with concussion is impossible because concussion is not a life-threatening disease [8]. As a result, a precise diagnosis of TAI in live patients with concussion can be limited. Since the 1960s, several studies have demonstrated TAI via autopsy in patients with concussion who presented no radiological evidence of brain injury [14,15,16]. In 1994, Blumbergs et al. detected TAI lesions via autopsy in the brain of five patients with concussion who died of other diseases [15]. However, the diagnosis of TAI in live patients with concussion was limited before the introduction of DTI because conventional brain MRI is not sensitive to the detection of TAI in concussion.

Since the 1980s, many researchers, including Povlishock, began to use the term “TAI” in their histopathological studies using animal brains [8,9,10,11,12]. Povlishock et al. observed axonal injuries upon a histopathological test of cat brains that experienced minor head injury via a fluid-percussion injury, but the cats did not show clinical abnormalities [9]. The authors assumed that the absence of clinical abnormalities in animals that exhibited axonal injuries in the brain might be because no more than one hundred to one hundred and fifty axons were affected, even though the corticospinal tract contains several hundred thousand axons in any animal [9]. Furthermore, they suggested that asymptomatic concussion patients might have a minor axonal injury and that post-concussion syndrome might be related to this type of axonal injury [9]. Other studies reported the possible presence of asymptomatic axonal injury in concussion or mTBI [10,11]. Consequently, Poblishock et al. reported that axonal injury is a common finding of all TBIs [10]. Moreover, the distribution and number of injured axons increase with the increasing severity of the trauma in mild, moderate, and severe TBI [12].

The history of the use of the term TAI with regard to the term “diffuse axonal injury” has caused some confusion in the clinical field [8,13]. Adams et al. (1982) began to use the term “diffuse axonal injury”, which defined microscopic axonal injuries in the white matter of the cerebral hemisphere, corpus callosum, and brainstem caused by mechanical forces during TBI [8,13,55]. Instead of diffuse axonal injury, “TAI” or “diffuse TAI” was used to correct the confusion in the term “diffuse”, because the location of the lesions of diffuse axonal injury was not diffuse but multifocal, and diffuse did not include the meaning of trauma as the etiology of axonal injury [3,7,8,13,56,57]. Generally, the traditional definition of diffuse axonal injury refers the patients with severe and prolonged coma at the onset of head trauma with poor outcomes [3,7,8,13,56,57]. The term “TAI” began to be used for these injuries instead of diffuse axonal injury because more limited patterns of axonal injury than classical diffuse axonal injury were observed in milder TBI with the introduction of DTI [3,7,8,13,56,57].

## 6. Detection of Traumatic Axonal Injury in Concussion Using Diffusion Tensor Imaging

After the development of DTI in the 1990s, several animal studies demonstrated TAI lesions in DTI consistent with the results of histopathological studies [58,59,60,61,62]. In 2007, Mac Donald et al. reported consistent TAI lesions in DTI and histopathological tests in a mouse brain that sustained a moderate cortical controlled-impact injury and showed normal findings in conventional MRI [58]. The authors concluded that DTI is more sensitive and suitable for detecting TAI than conventional brain MRI [58]. In addition, several other animal studies have also demonstrated the agreement of TAI lesions in DTI with a histopathological test in TBI brain models, including concussion and mTBI [59,60,61,62].

In 2002, Arfanakis et al. reported TAI in patients with mTBI using DTI [17]. Subsequently, several hundred DTI-based studies have detected TAI in patients with concussion or mild TBI [8,18,19,20,21,22,23,24,25,26,27,28,29,30,31]. Two methods were used to detect TAI in the above DTI-based studies: (1) the DTI region of interest (ROI) method, entailing the measurement of DTI parameters in specific ROIs of the brain; (2) the diffusion tensor tractography (DTT) method for neural tracts [8,29]. The DTI ROI method can result in false results because of the individual variability in the anatomical location of a neural structure [8,29,63]. The main advantage of DTT over DTI is that it can evaluate an entire neural tract using the DTT parameters and configurational abnormal findings, such as tearing, narrowing, or discontinuation (Figure 1) [8,29]. Furthermore, the DTT method has higher reliability than the DTI ROI method [64]. As a result, the DTT method appears to be more useful for detecting TAI in concussion than the DTI ROI method [8,29]. Several hundred papers observed TAI in patients with concussion or mTBI in various neural tracts, including the spinothalamic tract, corticospinal tract, fornix, and cingulum (Figure 2) [8,19,20,21,26,27,28,29,30,31].

After the introduction of DTI, dozens of studies have reported TAI in individual patients with concussion or mTBI. The majority of these studies demonstrated TAI using the DTT method [8,19,21,27,28,29,30,31]. On the other hand, a diagnosis of TAI cannot be based solely on DTI results because of the possibilities of asymptomatic axonal injury from previous head trauma, the presence of age-related changes, and the existence of analytic errors or artifacts related to DTI. According to recent studies on the diagnosis of TAI in concussion, clinicians can diagnose TAI after considering the patient’s prior medical history, head trauma situation, clinical features after head trauma, and the findings of physical examination, DTI, and results of other brain evaluation methods including nuclear medicine imaging and neurophysiological tests [8,51,52,53,54]. Additionally, improvement of a clinical symptom with the management of an injured neural tract could be additional evidence for TAI. For example, when a patient develops central pain due to injury of the spinothalamic tract following concussion, if the patient’s pain improves with the administration of specific drugs for central pain, that would be additional evidence for TAI in this patient [8]. Thus, DTI results can provide valuable data for the diagnosis of TAI in concussion. TAI lesions in concussion or mTBI can persist for approximately 10 years after the injury [65]. Therefore, DTI results are expected to provide useful data for diagnosing axonal injury in post-concussion syndrome [65].

## 7. Conclusions

This study reviewed TAI in patients with concussion. Concussion refers to transient changes in the neurological function of the brain resulting from head trauma. Therefore, it should not involve any organic brain injury [1,2,3]. However, TAI has been reported in autopsy studies of the human brain, histopathological studies of the animal brain, and conventional brain MRI following concussion before the development of DTI [14,15,16,34,36,37]. Furthermore, Poblishock et al. insisted that TAI is a common finding of all TBIs, including concussion and mTBI [12]. Nevertheless, the diagnosis of TAI in live patients with concussion had been limited because of the low resolution of conventional brain MRI [6,8]. Since the introduction of DTI in 2002, several hundred studies have reported TAI in patients with concussion [8,17,18,19,20,21,22,23,24,25,26,27,28,29,30,31]. Furthermore, dozens of studies have reported TAI using DTT for various neural tracts in individual patients with concussion [8,19,21,27,28,29,30,31]. As a result, DTI provides useful data for diagnosing TAI in patients with concussion. The confirmation of TAI in live patients with concussion can be limited because of the need for a confirmatory histopathological study by brain biopsy. As a result, further studies on the diagnostic criteria for TAI using DTI without histopathological tests in individual patients with concussion will be needed in the clinical field, even though a few recent studies suggested diagnostic approach methods for TAI in individual patients with concussion [29,30].

## Figures and Tables

**Figure 1 healthcare-10-00931-f001:**
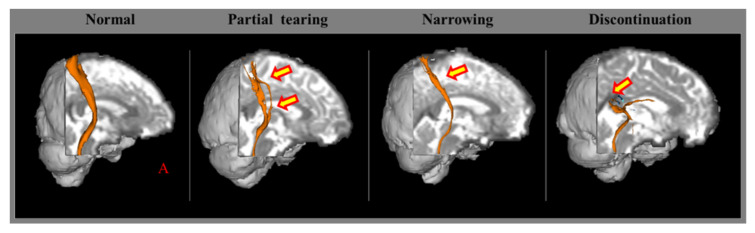
Configurational analysis of the spinothalamic tract in patients with mild traumatic brain injury (reprinted with permission from Jang, S.H., Traumatic Brain Injury. *In Tech*. 2018; 137–154) [7,16]. A: Anterior.

**Figure 2 healthcare-10-00931-f002:**
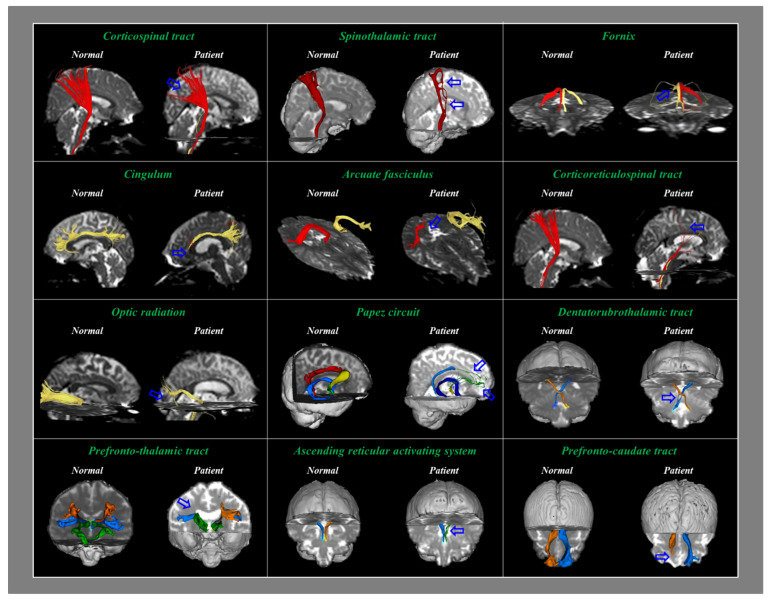
Traumatic axonal injuries of various neural tracts in patients with concussion or mild traumatic brain injury (reprinted with permission from *Brain & Neurorehabilitaiton*. 2016; 9(2): e1) [13].

**Table 1 healthcare-10-00931-t001:** Traumatic brain injury subtypes.

Patho-Anatomy			
Diffuse		Focal	
Concussion		Contusion	
Traumatic axonal injury/diffuse axonal injury		Penetrating	
Blast		Hematoma	
Abusive head trauma		Epidural	
		Subarachnoid	
		Subdural	
		Intraventricular	
		Intracerebral	
Severity of head trauma	LOC	PTA	GCS
Mild:	≤30 min	≤24 h	13~15
Moderate:	>30 min, ≤24 h	>24 h, ≤7 days	9~12
Severe:	>24 h	>7 days	3~8

LOC: loss of consciousness, PTA: post-traumatic amnesia, GCS: Glasgow coma scale (reprinted with permission from Jang, S.H., Traumatic Brain Injury. *In Tech*. 2018; 137–154) [1,6,7].

## Data Availability

Not applicable.

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
