# Peer review of "Hidden Truth in Cerebral Concussion—Traumatic Axonal Injury: A Narrative Mini-Review"

_healthcare, 2022, doi:10.3390/healthcare10050931_

Round 1

Reviewer 1 Report

Dear Editor
Thanks a lot for inviting me as the reviewer of this manuscript in which the authors have tried to assess the role of DTI in evaluation of patients with concussion. I read the manuscript carefully and tried to review it critically. I appreciate the effort put to this job by authors as they have addressed an important topic in the neurotrauma science. Although I think this is a valuable review of the topic, but it seems to add no serious updates to the literature. Review articles must pave the way for future original researches and contain drawbacks in the field. As the authors themselves have mentioned, several hundred studies have evaluated the role of DTI in diagnosis of concussion and its prognostic role for PCS. So, apart from the appropriate flow of the manuscript, it has no significant objective for the topic. The decision to accept the manuscript is related to the priorities of the journal.

Best regards,

Reviewer

Author Response

Answer:  Thank you for the reviewer’s comment. This manuscript is a narrative review for TAI in cerebral concussion. Because it is not a review for previous studies, we could not describe the strengths and weaknesses of each published studies. However, we described the limitation  of the DTI which is the main diagnostic tool for TAI in concussion and further studies as follows (underline).

Conclusions

This study reviewed TAI in patients with concussion. Concussion refers to transient changes in the neurological function of the brain resulting from head trauma. Therefore, it should not involve any organic brain injury [1-3]. On the other hand, TAI has been reported in autopsy studies of the human brain, histopathological studies of the animal brain, and conventional brain MRI following concussion before the development of DTI [12-14,32,35,36]. Furthermore, Poblishock et al. insisted that TAI is a common finding of all TBI, including concusion and mTBI [10]. Nevertheless, the diagnosis of TAI in live patients with concussion had been limited because of the low resolution of conventional brain MRI [6,33]. Since the introduction of DTI in 2002, several hundred studies have reported TAI in patients with concussion [6,15-29]. Furthermore, dozen of studies have reported TAI using DTT for various neural tracts in individual patients with concussion [6,17,19,25-29]. As a result, DTI provides useful data for diagnosing TAI in patients with concussion. The confirmation of TAI in live patients with concussion can be limited because of the need for a confirmatory histopathological study by brain biopsy. As a result, further studies on the diagnostic criteria for TAI using DTI without histopathological tests in individual patients with concussion will be needed in the clinical field, even though a recent few studies suggested diagnostic approach methods for TAI in individual patients with concussion [27,28].

Reviewer 2 Report

The authors review traumatic axonal injury (TAI) in concussion with history of TAI, limitation of conventional brain MRI, and clinical value of  DTI over MRI in concussion. The manuscript looks very interesting. However, I have some concerns which need to be addressed.

  1. Discuss more about future direction or strategy for TAI diagnosis in concussion.
  2. Please also discuss the current situation of drug development for TAI treatment in concussion. For example: Specific drugs which are used for current clinical trials or animal study in TAI treatment.
  3. Please revise the repeated clause “on the other hand”.
  4. In line pg.4; line 144, Please revised “Demonstration” with other word (e.g., Detection).

Author Response

Point 1)

Discuss more about future direction or strategy for TAI diagnosis in concussion.

Answer: We totally agree with the reviewer’s comment. We think that the estabilishment for diagnostic criteria for TAI in concussion is necessary in the near future first of all. So, we already described as follows in the conclusion (underline).

  1. Conclusions

This study reviewed TAI in patients with concussion. Concussion refers to transient changes in the neurological function of the brain resulting from head trauma. Therefore, it should not involve any organic brain injury [1-3]. However, TAI has been reported in autopsy studies of the human brain, histopathological studies of the animal brain, and conventional brain MRI following concussion before the development of DTI [12-14,32,35,36]. Furthermore, Poblishock et al. insisted that TAI is a common finding of all TBI, including concusion and mTBI [10]. Nevertheless, the diagnosis of TAI in live patients with concussion had been limited because of the low resolution of conventional brain MRI [6,33]. Since the introduction of DTI in 2002, several hundred studies have reported TAI in patients with concussion [6,15-29]. Furthermore, dozen of studies have reported TAI using DTT for various neural tracts in individual patients with concussion [6,17,19,25-29]. As a result, DTI provides useful data for diagnosing TAI in patients with concussion. The confirmation of TAI in live patients with concussion can be limited because of the need for a confirmatory histopathological study by brain biopsy. As a result, further studies on the diagnostic criteria for TAI using DTI without histopathological tests in individual patients with concussion will be needed in the clinical field, even though a recent few studies suggested diagnostic approach methods for TAI in individual patients with concussion [27,28].

Point 2)

Please also discuss the current situation of drug development for TAI treatment in concussion. For example: Specific drugs which are used for current clinical trials or animal study in TAI treatment.

Answer: We totally agree with the reviewer’s comment. As far as we are aware, there has been no specific drugs for treatment of TAI. The main principle of management for TAI is symptomatic management for each clinical feature which is developed after concussion. So, we revised as follows.

  1. Detection of traumatic axonal injury in concussion using diffusion tensor imaging

After the introduction of DTI, dozen of studies have reported TAI in individual patients with concussion or mTBI. The majority of these studies demonstrated TAI using the DTT method [6,17,19,25-29]. On the other hand, a diagnosis of TAI cannot be based solely on the DTI results because of the possibilities of asymptomatic axonal injury from previous head trauma, the presence of age-related changes, and the existence of analytic errors or artifacts related to DTI. According to recent studies on the diagnosis of TAI in concussion, clinicians can diagnose TAI after considering the patient’s prior medical history, head trauma situation, clinical features after head trauma, and the findings of physical examination, DTI, and results of other brain evaluation methods including nuclear medicine imagings and neurophysiological tests [6,50-53]. Additionally, improvement of a clinical symptom with management of an injured neural tract could be additional evidence for TAI. For example, when a patient develops central pain due to injury of the spinothalamic tract following concussion, if the patient’s pain improves with the administration of specific drugs for central pain, that would be additional evidence for TAI in this patient [6]. Thus, DTI results can provide valuable data for the diagnosis of TAI in concussion. TAI lesions in concussion or mTBI can persist for approximately 10 years after the injury [65]. Therefore, DTI results are expected to provide useful data for diagnosing axonal injury in post-concussion syndrome [65].

Point 3)

Please revise the repeated clause “on the other hand”.

Answer: We totally agree with the reviewer’s comment. So, we revised through the whole manuscript.

Point 4)

In line pg.4; line 144, Please revised “Demonstration” with other word (e.g., Detection).

Answer: We totally agree with the reviewer’s comment. So, we revised as follows.

6. Detection of traumatic axonal injury in concussion using diffusion tensor imaging

Reviewer 3 Report

The manuscript presented by Sung Ho Jang and Dong Hyun Byun, entitled "Hidden Truth in Cerebral Concussion-Traumatic Axonal Injury: A Narrative Mini-Review", is  original comparing with previous studies present in literature.

The authors show in detail how it has been developed the diffusion tensor imaging (DTI) approach.

The figures are clear and the manuscript is well written.

I found this review really interesting and suitable to be published in Healthcare journal.

Author Response

Answer: Thank you for the reviewer’s comment.

Round 2

Reviewer 2 Report

Please fix some spacing between words and reference numbers.